# Perceptions and Expectations of Academic Staff in Bucharest towards the COVID-19 Pandemic Impact on Dental Education

**DOI:** 10.3390/ijerph20031782

**Published:** 2023-01-18

**Authors:** Laura Iosif, Ana Maria Cristina Țâncu, Andreea Cristiana Didilescu, Marina Imre, Silviu Mirel Pițuru, Ecaterina Ionescu, Viorel Jinga

**Affiliations:** 1Department of Prosthodontics, Faculty of Dentistry, Carol Davila University of Medicine and Pharmacy, 17-21 Calea Plevnei Street, Sector 1, 010221 Bucharest, Romania; 2Department of Embryology, Faculty of Dentistry, Carol Davila University of Medicine and Pharmacy, 17-21 Calea Plevnei Street, Sector 1, 010221 Bucharest, Romania; 3Department of Professional Organization and Medical Legislation-Malpractice, Faculty of Dentistry, Carol Davila University of Medicine and Pharmacy, 17-21 Calea Plevnei Street, Sector 1, 010221 Bucharest, Romania; 4Department of Orthodontics and Dento-Facial Orthopedics, Faculty of Dentistry, Carol Davila University of Medicine and Pharmacy, 17-21 Calea Plevnei Street, Sector 1, 010221 Bucharest, Romania; 5Department of Urology, “Prof. Dr. Theodor Burghele” Clinical Hospital, Faculty of Medicine, Carol Davila University of Medicine and Pharmacy, Șoseaua Panduri 20, Sector 5, 050653 Bucharest, Romania

**Keywords:** dental education, dental teachers, COVID-19 pandemic, teaching performance, manual skills, digital education, virtual dentistry

## Abstract

Dental education was severely challenged by the COVID-19 pandemic worldwide. The evaluation of the viewpoint of the dental teachers of the Faculty of Dentistry at “Carol Davila” University of Medicine and Pharmacy in Bucharest, Romania, on these exceptional circumstances’ consequences was the objective of this paper. A cross-sectional study was conducted in April 2022, on the academic staff who reported their perceptions of the emotional and educational impact of the pandemic by completing a Google Forms questionnaire. Although a significant emotional impact of the pandemic was reported by over a third of the participants (31.2%), most of them being teachers of fifth-year dental students (*p* = 0.019), the perceived stress had an impact on the teaching performance in few of them (14%), the quality of sleep remaining unaffected in most of them (53.7%), whereas the level of anxiety was low (57%). An educational impact regarding the techno difficulties during the online transition was mentioned by few respondents (16.1%), with male teaching staff facing the fewest problems (*p* = 0.024), as well as low levels of difficulties in transmitting academic information (11.9), with men also being the most unaffected (*p* = 0.006). More than half of the participants (59.1%) rather see digital and/or virtual education during the pandemic as having adverse effects on the educational system, the most sceptical being teachers of the fifth (*p* = 0.001) and sixth years (*p* = 0.001). The COVID-19 pandemic affected the academic staff of the Faculty of Dentistry at “Carol Davila” University of Medicine and Pharmacy in Bucharest, Romania, not only at a personal level but also at a professional, pedagogical one, due to the introduction of the online teaching system followed by the hybrid one. Age group, gender, and teaching year differentiated the degree of emotional and educational impairment.

## 1. Introduction

Discovered in December 2019, a nanosized viral particle with an average genomic length of 30 kilobases [1], categorized in the Coronaviridae family as SARS-CoV-2, has suddenly and most unexpectedly shut down the entire world. Its spread with astonishing speed across continents, although initially predicted to be a potential epidemic with localization in China, was made possible by the sheer novelty of the virus—general immunity being virtually non-existent. Currently accounting for a total of 6,645,812 deaths, the COVID-19 (acronym for Coronavirus Disease 2019) pandemic ranks fourth in terms of pandemic mortality in our era, surpassing, in terms of years and the number of waves, historical pandemics such as the Spanish flu (1918–1919).

Globally, everything had to be reconfigured without any pre-determined pattern, the crisis solution for most nations being a lockdown. Its impact, only partially foreseeable, has hit specific social, economic, health and industrial sectors, one of the most difficult to predict in terms of its long-term effects and evolution being the education sector, despite its greater resilience than other sectors. In terms of resilience, the issue here is the process of digitisation of education systems, debated and implemented in many developed countries even many years before the onset of the COVID-19 pandemic, yet accelerated in an extraordinarily swift and professional way by millions of teachers and members of administrative and technical bodies in education systems around the world. The vast majority of them, completely unfamiliar with the use of digital platforms, tried and succeeded in just a few weeks to transfer their entire teaching activity online and to become familiar with the operation of digital platforms such as Moodle, Zoom, Google Meet, Udemy, WebEx, Microsoft Teams, etc. In spite of the extraordinary effort made, the university medical education system has felt the cracks of this e-learning and e-teaching model the most, in terms of the discontinuation of students’ direct practice [2] in university hospital units or in university dental clinics. The absence of direct clinical examination, the lack of real interaction with the patient and the so-called “virtual” practical therapeutic manoeuvres have had an impact on the students’ ability to take the initiative in making diagnoses, applying therapeutic decisions and on manual skills acquired through the development of superior psychomotor, communication and cognitive skills, especially in the case of dental students in their clinical semesters [3].

The discontinuation of traditional university dental education, involving the physical presence of students, with the introduction of lockdown measures according to the incidence of COVID-19 cases, has thus been carried out with very few exceptions in the individual states on each continent. As far as Europe is concerned, the COVID-19 pandemic has caused disruption and uncertainty in the treatment of patients in private dental practices and has disrupted teaching in most university centres, including the one in Bucharest, Romania.

As far as the curricula for bachelor studies at the Faculty of Dentistry of the “Carol Davila” University of Medicine and Pharmacy in Bucharest are concerned, it is structured over 12 semesters, the first to the sixth semester including mainly courses of fundamental, complementary and preclinical speciality subjects, whereas clinical speciality subjects are mostly introduced starting with the seventh semester and until the end of the twelfth semester. Before the COVID-19 pandemic, the teaching of all course forms was primarily delivered in person. However, this was completely stopped and switched to online mode with the onset of the pandemic and the declaration of a national state of emergency in Romania by Military Ordinance in March 2020. The new academic year started in October 2020 and was marked by the reopening of the Faculty of Dentistry in Bucharest by conducting clinical training activities in a hybrid system, i.e., both with the physical presence of students and also online. In 2022, we witnessed a full return of dental education activities to the established physical format, but due to the unpredictability of the risk of potential future pandemics and subsequent public health policies, the development of flexible curricula to include online learning, the establishment of an effective framework to identify barriers, limitations and vulnerabilities in the training of future dentists and to facilitate optimal solutions for them, remains a compelling necessity.

Correctly identifying this necessity, the scientific literature has analyzed massively and “on the fly” the impact of the COVID-19 pandemic on dental medical education systems on a global level, with multiple studies focusing first and foremost on the perception of students on the efficiency of online education, on psycho-emotional barriers as a result of social and financial constraints imposed by the pandemic, etc.

Previous studies have focused on the perceptions of university staff on the impact of the pandemic on general medicine teaching, with important results being reported [4,5,6]. Similar information was collected both from the dental medicine academic staff and dentistry students [7,8,9]. However, fewer studies have focused on dental teachers as a single target group on how the sudden transition from physical to online teaching was managed and perceived, the technological difficulties encountered, possible advantages and envisioned future solutions by them [10,11].

As far as our country Romania is concerned, the evaluation of the teaching staff in dentistry, addressing the same issue, was almost completely absent. Precisely for this reason, the authors of this manuscript started a study aimed at quantifying the impact caused by the changes that occurred in dental education during the pandemic from the perspective of academic teaching staff from the Faculty of Dentistry of the “Carol Davila” University of Medicine and Pharmacy in Bucharest, as well as the emotional one with a potential effect on teaching performance during and after the COVID-19 pandemic. In addition, we intended to identify the teaching staff groups most affected by the changes during this period.

## 2. Materials and Methods

### 2.1. Study Design and Sampling Procedures

The cross-sectional observational study was conducted on a representative group of teachers of the Faculty of Dentistry of the “Carol Davila” University of Medicine and Pharmacy in Bucharest, Romania. In order to evaluate the perspective of the academic teaching staff on the psychological barriers and the self-perceived impact regarding the quality of the educational activity experienced during the COVID-19 pandemic, all the academic teachers of the Faculty of Dentistry of the “Carol Davila” University of Medicine and Pharmacy in Bucharest (*n* = 320; females = 207, males = 113) teaching in the first to third (*n* = 132) and fourth to sixth (*n* = 188) academic years were invited via institutional emails on 4.04.2022 to participate anonymously in completing a questionnaire. From these, 93 university teachers (females = 66, males = 27) complied with our request, forming the present study group.

### 2.2. Data Collection and Ethical Considerations

The initial version of the questionnaire, carried out by three members of the Faculty of Dentistry, of the Carol Davila University of Medicine and Pharmacy in Bucharest, Romania was distributed in order to be evaluated by the other authors of the study, who were also teachers actively involved in the dental education system in Bucharest. The pilot questionnaire was reviewed in terms of the clarity of the wording and the questions were checked in order to cover all the study goals. The final version as well as the study protocol were approved by the Ethics Commission of the Scientific Research of the Carol Davila University of Medicine and Pharmacy in Bucharest, Romania, with the corresponding ethical approval no. 6122/08.03.2022. On 4 April 2022, the electronic questionnaire was uploaded to Google Forms (Alphabet Co., Mountain View, CA, USA), with the prior exclusion of duplicate answers from the platform settings and distributed, via their institutional addresses, to all teachers of the “Carol Davila” Faculty of Dentistry of the University of Medicine and Pharmacy in Bucharest, Romania. In the questionnaire header, the respondents were informed of their voluntary and anonymous participation, the purpose of the study and the estimated duration of the questionnaire completion. No incentives were used for study participation.

The questionnaire was carried out over 2 consecutive weeks, without issuing reminders to the potential respondents, precisely in order to be able to assess as objectively as possible the concern of the academic body for this subject, as previously mentioned to be one of the goals of our study. The questionnaire was accessible 24/7, and the questionnaire closed on 28 April 2022. Questionnaire submission by the academic teachers was considered as informed consent from their side to participate in this study. The answers were collected automatically and the institutional email addresses of the respondents were not recorded in the response sheet. During the data collection timeframe, no notifications regarding the technical functionality of the questionnaire were registered.

### 2.3. Survey Instrument

Given the absence of a validated instrument on this topic in the scientific literature, our questionnaire was constructed from scratch in Romanian, containing a set of 17 questions. These were divided into 3 sections, thus, the questions from Q1 to Q6 collecting mainly socio-demographic descriptive variables, questions Q7 to Q10 referring to the psychological impact felt by academics during the COVID-19 pandemic, and questions Q11 to Q17 referring to the impact felt in terms of dental education in the same pandemic context. The questions, their used abbreviations, the respective five-answer scale, as well as the Romanian translation, can be accessed in the Appendix A.

As a whole, the survey covered issues experienced by university teachers during the COVID-19 outbreak, such as emotional distress, the role of stress on professional performance, sleep quality, anxiety levels, and the technological barriers faced during the transition to online teaching, the difficulty and quality of teaching via digital platforms, the online assessment of students, the role of traditional simulators in the acquisition of practical manual skills by dental students on the one hand, and of revolutionary state-of-the-art simulation systems such as haptic devices or virtual reality on the other hand.

The questions in sections 2 and 3 (11 questions) included a scale with 5 answers, divided into ranks, with identical structure and of closed type, following the Likert scale model. The answer direction of the scale was randomly interspersed from one question to the next, this being from positive to negative for some questions and from negative to positive for others, in order to avoid the symmetric direction of the survey responses, and the potential survey bias.

Lastly, it should be mentioned that for a better assessment of the results obtained per question, we have considered it useful to apply a score, as follows: a score of 1 point was given for the “Strongly Disagree” option, 2 points for “Disagree”, 3 points for “Neutral”, 4 points for “Agree” and 5 points for “Strongly Agree”, regardless of the question answering direction of the scale.

A reliability analysis was performed on the questionnaire’s items, for the two categories: the psychological impact (questions 7 to 10) and the educational impact (questions 11 to 17) of the COVID-19 pandemic. The Cronbach’s alpha coefficient for the two categories was close to or above 0.7. More precisely, the items which analyzed the psychological impact had an internal consistency of α = 0.686 and the items which analyzed the educational impact had an internal consistency of α = 0.767.

### 2.4. Data Management and Analysis

Dental teachers’ answers were automatically collected into Google Forms. A data sheet was generated in Microsoft Excel (Microsoft Corporation, Redmont, Washington, WA, USA), in which the variables were coded. Data were thereafter transferred and analysed using IBM SPSS Statistics 25 (IBM Corp. Released 2017. IBM SPSS Statistics for Windows, Version 25.0. Armonk, NY, USA: IBM Corp.) and illustrated using Microsoft Office Excel/Word 2013 (Microsoft Corporation, Redmont, Washington, WA, USA). Quantitative variables were tested for normal distribution using the Shapiro–Wilk test and were written as averages with standard deviations or medians with interquartile ranges. Quantitative independent variables with non-parametric distribution were tested between groups using Mann–Whitney U test. Qualitative variables were written as counts or percentages and differences between groups were tested using Fisher’s exact test. Z-tests with Bonferroni correction were used to further detail the results obtained from the contingency tables.

## 3. Results


**Section 1—Socio-demographic data**


The response rate of the faculty teachers was 29.06% (*n* = 93). Data from Table 1 show the demographic characteristics of the university teachers. Most of them had an age between 45–60 years (52.7%) or 25–44 years (41.9%), most of them being women (71%). 23.7% of the respondents were included in the category of teachers with more than 25 years in university dental education, while 22.6% were employed for 16–20 years. Many teachers from the study had multiple years of medical school for teaching, the most frequent years were the fifth year (39.8%), sixth year (28%), third year (25.8%) and first year (25.8%). According to the academic degree, only 14% were university professors, and most of them were assistant professors (52.7%). During the pandemic, 40.7% of the university teachers confirmed SARS-CoV-2 infection.

Data from Figure 1 show the distribution of the university teachers according to age and gender. Differences between groups are significant according to Fisher’s exact test (*p* = 0.031) and Z-tests with Bonferroni correction showed that university teachers with age between 25–44 years were more significantly men than women (63% vs. 33.3%) while university teachers with age between 45–60 years were more significantly women than men (60.6% vs. 33.3%).

The distribution of the university teachers according to employment duration and gender can be extrapolated from Figure 2. Differences between groups are significant according to Fisher’s exact test (*p* = 0.011) and Z-tests with Bonferroni correction showed that university teachers with an employment duration between 16–20 years were more significantly men than women (37% vs. 16.7%) while university teachers with an employment duration between 21–25 years were more significantly women than men (24.2% vs. 3.7%).

Regarding the distribution of the university teachers according to the answers from the survey, the following data were recorded:
**Section 2—Psychological impact (Table 2a)**-For question no. 7 (Emotionally affected), most of the personnel agreed (31.2%) or were neutral (29%). The median of the score (Score-Q7) was 3 points (IQR = 2–4 points) meaning a neutral state for this question;-For question no. 8 (Stress impact), most of the university teachers disagreed (39.8%) or strongly disagreed (32.3%). The median of the score (Score-Q8) was 2 points (IQR = 1–3 points) meaning a disagreement for this question;-For question no. 9 (Sleep quality), most of the personnel strongly disagreed (33.3%) or were neutral (26.9%). The median of the score (Score-Q9) was 2 points (IQR = 1–3 points) meaning a disagreement for this question;-For question no. 10 (Anxiety), most of the university teachers disagreed (35.5%) or strongly disagreed (21.5%). The median of the score (Score-Q10) was 2 points (IQR = 2–3 points) meaning a disagreement for this question.**Section 3—Educational impact (Table 2b)**-For question no. 11 (Technodifficulties), most of the respondents strongly agreed (37.6%) or agreed (32.3%). The median of the score (Score-Q11) was 4 points (IQR = 3–5 points) meaning an agreement for this question;-For question no. 12 (Difficulties in transmitting academic information), most of the university teachers disagreed (48.4%) or strongly disagreed (30.1%). The median of the score (Score-Q12) was 2 points (IQR = 1–2 points) meaning a disagreement for this question;-For question no. 13 (Quality of the academic information), most of the personnel agreed (30.1%) or disagreed (24.7%). The median of the score (Score-Q13) was 3 points (IQR = 2–4 points) meaning a neutral state for this question;-For question no. 14 (Difficulty of objective assessment of students), most of the personnel agreed (47.3%) or strongly agreed (30.1%). The median of the score (Score-Q14) was 4 points (IQR = 4–5 points) meaning an agreement for this question;-For question no. 15 (Acquiring practical manual skills by using traditional simulators), most of the personnel disagreed (31.2%) or agreed (26.9%). The median of the score (Score-Q15) was 3 points (IQR = 2–4 points) meaning a neutral state for this question;-For question no. 16 (Acquiring practical manual skills by using haptic devices or virtual reality), most of the personnel disagreed (37.6%) or were neutral (24.7%). The median of the score (Score-Q16) was 2 points (IQR = 2–3 points) meaning a disagreement for this question;-For question no. 17 (Effects of digital and/or virtual dental education), most of the personnel disagreed (38.7%) or were neutral (24.7%). The median of the score (Score-Q17) was 2 points (IQR = 2–3 points) meaning a disagreement for this question.**Analysis of the impact in accordance with gender/academic year**

Further to the analysis of the answers according to the respondents’ gender, the highest differences were obtained for Q11 and Q12, as shown in Table 3. According to the Mann–Whitney U tests differences between genders were statistically significant, data show that in university teachers, men agreed significantly more than women that the transition to online education occurred without technological difficulties (men: median score—5 points, IQR = 4–5 points vs. women: median score—4 points, IQR = 3–5 points, *p* = 0.024) and also men disagreed significantly more than women that the practice of transmitting academic information through digital platforms encountered major difficulties (men: median score—1 point, IQR = 1–2 points vs. women: median score—2 points, IQR = 2–3 points, *p* = 0.006).

The analyses by year of study showed that the largest intergroup differences were obtained for the first and fourth to sixth years for certain questions, as shall be shown below in Table 4. According to the results:-University teachers that teach to the first year (*p* = 0.045) or fourth year (*p* = 0.029) disagreed more while teachers that teach to the sixth year (*p* = 0.004) agreed more that the quality of academic information suffered because of the pandemic;-University teachers that teach to the fourth year (*p* = 0.050) agreed more while university teachers that teach to the sixth year (*p* = 0.028) disagreed more that acquiring practical manual skills can be successfully made using traditional simulators;-University teachers that teach to the first year (*p* = 0.040/*p* = 0.004) agreed more while university teachers that teach to the fifth year (*p* = 0.007/*p* = 0.001) or sixth year (*p* = 0.035/*p* = 0.001) disagreed more that virtual reality can be beneficial in acquiring practical manual skills/brought more favourable effects than disadvantages;-University teachers that teach to the fifth year agreed more that they were more emotionally affected (*p* = 0.019) or that they were having difficulties in transmitting academic information (*p* = 0.029) and also disagreed more about the stress impact of the pandemic (*p* = 0.022);-University teachers that teach to the sixth year agreed more that they were having difficulties in the objective assessment of students (*p* = 0.026).

## 4. Discussion

The COVID-19 pandemic, which has already spread over a period of more than two years, starting on 31.12.2019 and still ongoing today, is rightfully considered “as humanity’s worst crisis since World War II” [12]. Within university medical education in all countries around the world, it has generated unprecedented challenges and raised serious questions about the professional future for several generations of junior doctors, with simultaneous consequences at the level of their psychosomatic health [13,14]. Students’ medical practice carried out on patients in university clinics and hospitals during the pandemic has been severely disrupted, with major effects especially in dentistry, as an eminently practical branch [15]. Thus, vulnerabilities experienced by dental students such as psychological [16,17,18,19], or educational ones [20,21,22] as well as those related to the acquisition of psychomotor skills [23] with different reports depending on the year of study [21], as well as the stage of the COVID-19 pandemic when they were identified [24], were captured by the academic staff worldwide through web or internet surveys [17,25,26]. The response rate among dental students to these questionnaires was generally variable, reaching for example a high of 80% for the University of Giessen, Germany [27], 90.72% for the only dental school in Malta [24], 72% for the University of Jordan [15], but also low, as reported for example by the University of Washington, USA, of 35.5% [28], or moderate, such as that of the Faculty of Dental Medicine in Vienna, Austria, of 47% [29], and by our faculty in Bucharest, Romania with a percentage of 48.56% [30]. Although we would have expected that the problem of dental education in the era of COVID-19 would find a wider response among university teachers than among students in the same university in Romania, or at least close to the same, as reported by Jum’ah et al. [22], the response rate of 29.06% in our study was quite low. An explanation could be the fact that at the time of our survey, at least three other questionnaires on various educational topics had been launched addressing the teachers of the Faculty of Dentistry, all related to the period of the COVID-19 pandemic. Another explanation could be the overall reduced willingness of university teachers to respond to surveys that require using computer technology, comparative to the higher response using the traditional mailed survey, as previously reported in the literature [31,32], but also with the different timing of the student vs. academic staff questionnaire launches during the COVID-19 pandemic in Bucharest (5 December 2020 vs. 4 April 2022).

Our survey has unequivocally aroused a major interest among female dental teachers, which is in accordance with the worldwide gender distribution in medical studies [33], most of the respondents belong to the age group 45–60 years, with a career of more than a decade and a half in university dental education. This might be explained by the majority percentage of female dentistry teachers in the Faculty of Dentistry of the Carol Davila University of Medicine and Pharmacy in Bucharest, as reported in the Materials and Method section of this paper. Our findings are in agreement with other reports, such as those from Austria, where at least in an important speciality segment such as pedodontics, female specialists predominate, but also in general by the dynamically increasing trend of their presence in dental schools over the last decades [34]. Another remarkable characteristic of our respondents that needs to be discussed is the fact that most of the survey participants were academics teaching students in the final years of dentistry, i.e., the fifth and sixth year, a stronger concern among the same subgroup was also reported by Mukhatar K. et al. [35], who, however, included in their study members of the teaching staff from both dentistry and medicine. The concern of the academic staff arises, in our opinion, from the fact that for this group of students, whose usual practice with patients has been suddenly discontinued and then severely limited, ensuring an optimal level of clinical competence is questionable.

Regarding the answers to the psychological section consisting of four questions (Q7–Q10), almost half of the studied group reported emotional affect (Q7) due to the experience of the COVID-19 pandemic. A study with a large sample size previously performed in Spain obtained a similar percentage of faculty members with moderate or severe emotional impact scores, as compared to our results [36]. However, the stressful experience associated with the pandemic does not seem to extend to the online teaching performance of our respondents (Q8), at least at a personal perception level. The paucity of scientific literature on this topic is certain, although, in our opinion, this result is debatable, taking into account other factors, such as insufficient specialized training and support in e-teaching received by university professors from our target group prior to the sudden onset of the pandemic. It is already known that teachers who have not received professional training tend to teach online courses just the same as they deliver lectures in classical classrooms, omitting the differences between online teaching and face-to-face education [37].

On the other hand, the mainly unaffected quality of sleep (Q9) revealed by the teaching staff in our study still represents an argument in favour of maintaining a qualitative level of academic performance through online teaching methods that are insufficiently or even not previously pre-tested. Although coronasomnia [38], defined as sleep impairment during the COVID-19 pandemic, reached increased levels among healthcare workers worldwide, as pointed out previously [39], the late time of launching our questionnaire related to the onset of the pandemic, namely during the fifth Omicron wave, may be an explanation for this return

Further, in a Romanian study [40], the level of anxiety related to the SARS-CoV-2 infection investigated using the Fear of COVID-19 Scale was determined to be lower among dentists who are also academic staff, than in other studies conducted on general healthcare professionals or dentists without academic assignment [40,41,42,43,44]. Although we did not comparatively evaluate several professional groups as in the case above, our finding was similar, as most of the teachers in our study reported a rather low level of anxiety regarding the SARS-CoV-2 infection after returning to established on-site dental education. These findings may have several explanations, one of them being the time period in which our study was conducted, namely 16 months after the onset of the pandemic, and 12 months after the return to face-to-face teaching, albeit initially in a hybrid system. In this regard throughout this considerable period of time, the efficacy of vaccination had already a positive effect on the decrease in fear and anxiety levels, especially in dental professionals [45]. Another explanation for the reduced level of anxiety towards SARS-CoV-2 infection in our study group can be found in the international epidemiological context, since in Romania the number of cases and deaths caused by COVID-19 was not as high as in other countries, such as Spain, Italy, Israel, USA, South Africa, Germany, Turkey, England, France, etc. [46].

The educational section of our questionnaire including a set of 7 questions (Q11–Q17) was entirely focused on the teachers’ perception of the sudden shift to exclusively online, virtual teaching forced by the pandemic. Despite the breakthroughs in IoT (Internet of Things) that marked the year 2020, including currently in European universities and in the USA [47], the implementation of most systems based on virtual reality technologies still faces technological, learning, familiarization, funding and other challenges. The technological challenges to which we refer in Q11, encountered by some of the university teachers in our study do not, therefore, reveal an unexpected finding. It has been previously reported that most participants at eight universities in Malaysia were not fully comfortable with e-learning as a tool for teaching and this perception was attributed to technological challenges [48]. Thus, during the COVID-19 pandemic period, it has been recognized as “the phenomenon of technostress”, reported especially among teachers [49,50,51].

Moreover, another important finding of our study regarding technological difficulties was that female academics encountered the majority of them. Furthermore, female academics represented the group with the most significant difficulties in learning to transmit academic information through digital platforms, an issue investigated in our survey by Q12. Our findings are in agreement with Martin F. et al. [52] and Cassachia M. et al. [53], who recorded the same challenges in terms of technology and digital learning on the other side of the barricade, namely among female students. This gender digital gap among the academic staff in our study may have its origins in the socio-pedagogical concepts of the decades following the 1980s, after the introduction of the first PC (Personal Computer) by IBM (International Business Machines). These involved the preferential guidance of boys from the first stage of their early childhood towards the technological field, both by teachers and parents [54], and there are even reports of greater digital skills in boys [55,56,57]. Nevertheless, in our opinion, the ICTs (Information and Communication Technologies) are no longer a gender-stereotyped domain, Mc Adam M et al. [58] even use the terms digital emancipation and cyberfeminism for the current decade.

Further, regarding the quality of the transmission of academic information via digital platforms investigated by Q13, one aspect that drew our attention was that the greatest difficulties were encountered by teachers in the terminal years, belonging exclusively to clinical fields. For the university teachers who traditionally teach the clinical curriculum by means of a significant share of demonstrations of practical therapeutic maneuvers directly on the patient, as it is the case in our university, the pedagogical pandemic scenario has depreciated the clinical communication skills of the teachers and implicitly the quality of teaching the terminal year courses, an observation converging with that of Mukhtar K. et al. [35], which on a more limited group of faculty members even reports an inefficiency to teach psychomotor skills At the same time, the preclinical teachers and those teaching at the edge between clinical and preclinical fields—and here we specifically refer to some courses in our faculty in the fourth year of study, such as Fixed Prosthodontics—reported that the quality of teaching of the curriculum was well-maintained in the digital system, the reason being the fact that they were already familiarized with some e-learning systems which had been partly integrated before the pandemic onset.

The assessment of students is a pivotal stage of the educational process, for which we wanted to investigate the perception of academic dentistry teachers regarding the objectivity of online evaluations imposed by the pandemic, as requested by Q14. It is recognised that one of the main motivators for learning is their assessment, which most of them focus on as an indicator of their own performance [59]. A 3-year research project by Higgins R. et al. [60], investigating the meaning and impact of assessment feedback for students in higher education, showed that assessment is part of a constructivist theory of learning and starting from the benefits of assessment in the learning process, the lockdown period caused a major disruption in this regard. The assessment of the students of the Faculty of Dentistry of the Carol Davila University of Medicine and Pharmacy in Bucharest in the online system has also suffered, the most difficult being for the teachers who assessed students during the clinical years. This is in agreement with Hattar et al. [15], who also reported among final-year students the opinion that online assessment is not a good method for evaluation. Our questionnaire did not omit the issue of attempted fraud during assessments, which the shift to e-exams has amplified, as confirmed by our results, also in line with the reports of Mukhatar K. et al. [35], Egarter, S. et al. [61], and Chirumamilla A. et al. [62]. However, in the assessment carried out within three universities in Australia, it was surprisingly found by students that cheating was harder in online examinations than in traditional invigilated exams, a statement that contrasted with the perceptions of academic staff [63]. This points out the importance and legitimacy of our question regarding the objectivity of the assessment perceived by the academic staff, as it is obvious that clinical assessments cannot be carried out properly in a digital format.

The development of manual and cognitive skills during the training of future dentists has always been facilitated by the use and learning of therapeutic manoeuvres using traditional simulators, such as acrylic artificial dental fillings, extracted teeth, or phantom heads, the last invented in 1930 by Oswald Fergus [64] and known as “the gold standard in restorative dentistry” [65] for the satisfactory replication of the real oral environment, the positioning of the dentist in relation to the patient during the performance of different dental procedures. Consequently, the acquisition of practical skills by dental students through the exclusive use of traditional simulators was the topic for Q15 in our survey. As a result, they were not considered sufficient by the majority of our respondents, albeit according to a recent report [66], students’ performance through traditional simulation would be even superior to the haptic one. Furthermore, among the academic staff teaching in the clinical years in our study, the only ones who showed optimism in the integral acquisition of manual skills, by practicing dental procedures on traditional simulators by the students, were those from the fourth year. We explain the positive feedback from them by the fact that the fourth year is eminently a year of transition between virtual and real patient practice, between clinical and preclinical, and the patient practice is not as condensed and extensive as in the case of the final-year students. However, the reports are quite divergent on this topic, so, for example, Fugill [67], who determined that the clinical teachers‘ perception was that students do not learn complex clinical skills in the “phantom head” setting. Their arguments ranged from the lack of communication skills with patients, the work in the difficult conditions of the wet oral environment (saliva/blood), in the presence of perioral muscles and tongue strength and tone and many others. Today, modern dental simulators are based on virtual technologies, whose practical testing in the training of dental students started around 2004 [68] and whose role in increasing the quality of dental education, was confirmed by the most recent data in the scientific literature [69,70]. These innovative systems are based on three-dimensional (3D) environments, augmented reality (AR) systems, virtual reality (VR) and building information modelling [71,72,73,74]. Nonetheless, at the time of the outbreak of the COVID-19 pandemic in Romania, digital and haptic technologies had advanced modestly in the dental curricula, with variable degrees of integration in the academic pedagogical system, mainly testing, depending on the local resources and needs. They consisted of testing of virtual and augmented reality systems through the development in 2011 of the VirDenT system and its use at the Faculty of Dental Medicine in Constanta [75], or of virtual e-learning platforms based on 3D applications for dental prosthetics as a training method at the Faculty of Dental Medicine in Bucharest in 2016 [76].

The penultimate question of our questionnaire (Q16) was thus designed to evaluate whether faculty teachers considered realistic and optimal for developing practical manual skills by dental students the exclusive use of simulation systems, such as haptic devices or virtual reality supported by e-learning platforms during and after the COVID-19 pandemic. In this regard, prior to the COVID-19 pandemic, Steinberg et al. [77] emphasized enthusiastic feedback from the faculty members about its potential for developing basic procedural skills in students. Along the same line, experienced dental faculty members investigated by Gal et al. [78], also in the pre-pandemic, found that a newly developed haptic simulator could provide excellent benefits in the self-learning of manual dental skills by dental students. Other reports from dental teachers [2,79] during the pandemic pointed out high confidence in the efficiency of simulation exercises without the student’s physical presence in the clinical environment and without direct contact with patients, which contrasted with the perception of the majority of the teaching staff investigated by us.

It is interesting to point out that the same category of respondents who most strongly distrusted simulations systems regarding the acquisition of practical skills, namely academic teachers of terminal year undergraduate dental students, were also those who most appreciated the favourable effects of digital and/or virtual education for dentistry, according to the last item of our survey (Q17). Our finding can be interpreted as a vote of confidence for the future use of virtual resources along with traditional clinical practices on real patients, namely for a future dynamic hybrid strategy in dental education in Romania. Coming to an end, it should be mentioned that our study has several limitations. The first limitation of this study is the below-average response rate of university teachers to our questionnaire, which, although not necessarily so, may indicate the potential for bias in the results. This does not necessarily mean that bias exists, as there is, in our opinion, a possibility of no significant differences between the responses coming from those who responded to the survey and the way nonresponders would have responded had they taken the survey. Secondly, a generalization of the results may also be difficult, as the study was carried out in a single dental faculty in our country. A third limitation, as in any other study with a similar methodological structure, is the use of a questionnaire with a closed question format, which limited the possibility to provide free text responses and observations, the discovery of a broader perspective and thus possible solutions for the future of clinical teaching in dentistry. However, we consider that our study brings more information from an insufficiently investigated perspective, that of university teachers, regarding the forced educational paradigm shift in dentistry due to COVID-19 and from which new opportunities for reforming the Romanian dental school can start.

## 5. Conclusions

As the purpose of our study was to assess the viewpoint of the teachers of the Faculty of Dentistry of Carol Davila University of Medicine and Pharmacy in Bucharest, Romania, about the impact of the COVID-19 pandemic and the lockdown periods, along with the development of the undergraduate curriculum in an online and later hybrid system in the dental school, we can conclude that in our Faculty of Dentistry, it has had effects at a personal, psychological level, as well as professional, pedagogical consequences on the majority of the academic staff. Gender, age, as well as clinical teaching, are important parameters to be taken into consideration when judging the degree of emotional and educational impairment during emergency situations.

At the same time, with all its limitations, the study shows that the e-learning system and the virtual reality technologies that can be incorporated into it, can and should become at least an alternative form of teaching practical dental skills, even in the post-pandemic future. Long-term studies are needed to assess not only the psychological and emotional effects on academic staff but also the quality of teaching in dental sciences, in relation to pandemic-induced changes.

## Figures and Tables

**Figure 1 ijerph-20-01782-f001:**
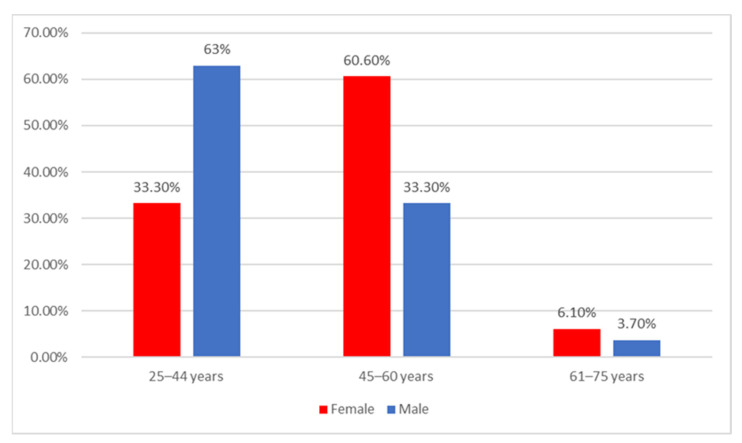
Distribution of the university teachers according to age and gender.

**Figure 2 ijerph-20-01782-f002:**
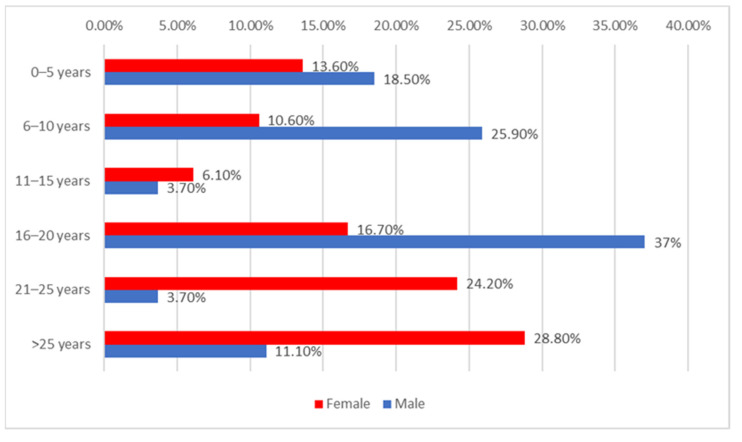
Distribution of the university teachers according to employment duration and gender.

**Table 1 ijerph-20-01782-t001:** Demographic characteristics of the university teachers.

Parameter (Nr., %)	Value
Q1—Age	39 (41.9%) 25–44 years, 49 (52.7%) 45–60 years, 5 (5.4%) 61–75 years
Q2—Gender	66 (71%) female, 27 (29%) male
Q3—Employment duration	14 (15.1%) 0–5 years, 14 (15.1%) 6–10 years, 5 (5.4%) 11–15 years, 21 (22.6%) 16–20 years, 17 (18.3%) 21–25 years, 22 (23.7%) > 25 years
Q4—Teaching university year of study	24 (25.8%) first year, 14 (15.1%) second year, 24 (25.8%) third year, 13 (14%) fourth year, 37 (39.8%) fifth year, 26 (28%) sixth year
Q5—Academic degree	49 (52.7%) assistant professors, 14 (15.1%) lecturers, 17 (18.3%) associate professors, 13 (14%) university professors
Q6—SARS-CoV-2 Infection	37 (40.7%)

**Table 2 ijerph-20-01782-t002:** (**a**) Distribution of the university teachers according to the answers from the survey, on the psychological impact. (**b**) Distribution of the university teachers according to the answers from the survey, on the educational impact.

Item /Answer	Strongly Disagree	Disagree	Neutral	Agree	Strongly Agree
(**a**)					
Q7	7 (7.5%)	23 (24.7%)	27 (29%)	29 (31.2%)	7 (7.5%)
Q8	30 (32.3%)	37 (39.8%)	13 (14%)	10 (10.8%)	3 (3.2%)
Q9	31 (33.3%)	19 (20.4%)	25 (26.9%)	13 (14%)	5 (5.4%)
Q10	20 (21.5%)	33 (35.5%)	18 (19.4%)	16 (17.2%)	6 (6.5%)
(**b**)					
Q11	3 (3.2%)	12 (12.9%)	13 (14.0%)	30 (32.3%)	35 (37.6%)
Q12	28 (30.1%)	45 (48.4%)	9 (9.7%)	9 (9.7%)	2 (2.2%)
Q13	14 (15.1%)	23 (24.7%)	14 (15.1%)	28 (30.1%)	14 (15.1%)
Q14	2 (2.2%)	7 (7.5%)	12 (12.9%)	44 (47.3%)	28 (30.1%)
Q15	9 (9.7%)	29 (31.2%)	20 (21.5%)	25 (26.9%)	10 (10.8%)
Q16	18 (19.4%)	35 (37.6%)	23 (24.7%)	14 (15.1%)	3 (3.2%)
Q17	19 (20.4%)	36 (38.7%)	23 (24.7%)	12 (12.9%)	3 (3.2%)

**Table 3 ijerph-20-01782-t003:** Comparison of Score-Q11 and Score-Q12 according to gender.

**Gender—Score-Q11**	**Average ± SD**	**Median (IQR)**	**Mean Rank**	***p* ***
Female (*p* < 0.001 **)	3.74 ± 1.11	4 (3–5)	43.14	0.024
Male (*p* < 0.001 **)	4.22 ± 1.18	5 (4–5)	56.43	
**Gender—Score-Q12**	**Average ± SD**	**Median (IQR)**	**Mean Rank**	***p* ***
Female (*p* < 0.001 **)	2.2 ± 0.96	2 (2–3)	51.52	0.006
Male (*p* < 0.001 **)	1.7 ± 0.99	1 (1–2)	35.94	

* Mann–Whitney U Test, ** Shapiro–Wilk Test.

**Table 4 ijerph-20-01782-t004:** Comparison of analysed scores according to the year of study.

First Year Dental Education					
Group/Score		Average ± SD	Median (IQR)	Mean Rank	*p* *
**Score-Q13**	Absent (*p* < 0.001 **)	3.22 ± 1.31	4 (2–4)	50.22	0.045
	Present (*p* = 0.009 **)	2.58 ± 1.28	2 (2–4)	37.73	
**Score-Q16**	Absent (*p* < 0.001 **)	2.32 ± 1.06	2 (1.5–3)	43.74	0.040
	Present (*p* = 0.009 **)	2.83 ± 1.00	3 (2–3)	56.38	
**Score-Q17**	Absent (*p* < 0.001 **)	2.2 ± 0.96	2 (1.5–3)	42.44	0.004
	Present (*p* = 0.067 **)	2.96 ± 1.12	3 (2–4)	60.10	
**Fourth Year Dental Education**					
**Score-Q13**	Absent (*p* < 0.001 **)	3.18 ± 1.31	3.5 (2–4)	49.39	0.029
	Present (*p* = 0.082 **)	2.31 ± 1.25	2 (1–3)	32.27	
**Score-Q15**	Absent (*p* < 0.001 **)	2.88 ± 1.15	3 (2–4)	44.86	0.050
	Present (*p* = 0.011 **)	3.62 ± 1.26	4 (2–5)	60.19	
**Fifth Year Dental Education**					
**Score Q7**	Absent (*p* < 0.001 **)	2.86 ± 1.06	3 (2–4)	41.86	0.019
	Present (*p* = 0.002 **)	3.38 ± 1.03	4 (3–4)	54.78	
**Score Q8**	Absent (*p* < 0.001 **)	2.34 ± 1.14	2 (1.25–3)	51.96	0.022
	Present (*p* < 0.001 **)	1.81 ± 0.90	2 (1–2)	39.50	
**Score Q12**	Absent (*p* < 0.001 **)	1.88 ± 0.91	2 (1–2)	42.39	0.029
	Present (*p* < 0.001 **)	2.32 ± 1.05	2 (2–3)	53.97	
**Score Q16**	Absent (*p* < 0.001 **)	2.7 ± 1.06	3 (2–3)	52.93	0.007
	Present (*p* < 0.001 **)	2.08 ± 0.98	2 (1–3)	38.03	
**Score Q17**	Absent (*p* < 0.001 **)	2.7 ± 1.07	3 (2–3)	54.33	0.001
	Present (*p* < 0.001 **)	1.95 ± 0.84	2 (1–2)	35.91	
**Sixth Year Dental Education**					
**Score Q13**	Absent (*p* < 0.001 **)	2.81 ± 1.32	3 (2–4)	42.10	0.004
	Present (*p* = 0.002 **)	3.69 ± 1.12	4 (3–4.25)	59.62	
**Score Q14**	Absent (*p* < 0.001 **)	3.81 ± 1.04	4 (3–5)	43.40	0.026
	Present (*p* < 0.001 **)	4.35 ± 0.56	4 (4–5)	56.27	
**Score Q15**	Absent (*p* < 0.001 **)	3.15 ± 1.15	3 (2–4)	50.71	0.028
	Present (*p* = 0.007 **)	2.54 ± 1.17	2 (2–4)	37.44	
**Score Q16**	Absent (*p* < 0.001 **)	2.6 ± 1.06	2 (2–3)	50.54	0.035
	Present (*p* = 0.001 **)	2.08 ± 1.01	2 (1–3)	37.88	
**Score Q17**	Absent (*p* < 0.001 **)	2.61 ± 1.07	3 (2–3)	52.36	0.001
	Present (*p* < 0.001 **)	1.85 ± 0.78	2 (1–2)	33.19	

* Mann–Whitney U Test, ** Shapiro–Wilk Test.

## Data Availability

The data presented in this study are available from the corresponding authors upon reasonable request.

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
