# Peer review of "Perceptions and Expectations of Academic Staff in Bucharest towards the COVID-19 Pandemic Impact on Dental Education"

_ijerph, 2023, doi:10.3390/ijerph20031782_

Round 1

Reviewer 1 Report

Dear authors, the manuscript has been reviewed and below are the comments. I hope these comments would help to improve the current manuscript.

Abstract:

-          The abstract is quite lengthy. Please summarize it with pertinent information on the psychological impact felt by academics during the COVID-19 pandemic, and the impact felt in terms of dental education in the same pandemic context, rather than listing out the entire results.

-          However, to the best of our knowledge, no scientific study has focused strictly on the vision of the academic body in the field of dentistry ….” This phrase may not be appropriate as currently there is much evidence, in fact several review articles have been published on this matter.

Introduction:

-          Nonetheless, to the best of our knowledge, there is no study described in the literature that strictly investigated the perspective of the dentistry academic body on the way the abrupt transition from physical to online…” Similarly, please revise this sentence. Perhaps, such a study may be considered novel in a particular country, such as Romania, but it is not considered novel throughout the globe as similar studies have been conducted in several countries.

Methodology:

-          93 university teachers (n=93) complied with our request, forming the present study…” Can omit the (n=93) since the authors have listed the number of teachers who participated in the study.

-          Please summarize “2.3. Survey Instrument”.

-          Did the authors perform any reliability test on the questionnaire items such as Cronbach alpha?

Results:

-          I would highly suggest rewriting the results. It is advisable to categorize the results in a manner that suit the questionnaire design. The authors designed the questionnaire into three sections. Hence, present the results accordingly. For instance, the authors classify section 2 as psychological impact. So, how are the overall results of the psychological impact? Is it the majority of the teachers agreed that their psychological impacts are significant due to the pandemic? Besides, the authors divided the psychological impact into emotionally affected, stress impact, sleep quality and anxiety. Which one has the most significant impact?

-          The readers can understand based on the tables and thus, the authors should summarize the results instead of repeating what is written in the tables.

-          There are lots of tables which some of them can be merged.

Discussion:

-          Within the university medical education in all countries around the world it has generated …family life, as well as on their professional prospects, which have been severely affected by the prolonged lockdown.” Please at least put a reference to these statements.

-          Some of the discussion points are not relevant to the findings. The discussion should be confine to what happened to the findings? What factors could lead to the present findings? Are the findings in accordance or contradicts previously similar studies? What recommendations the authors would like to propose?

Author Response

Dear Reviewer,

Thank you for giving us the opportunity to submit a revised form of the manuscript Perceptions and Expectations of University Teachers towards the COVID-19 Pandemic Impact on Dental Education authored by Laura Iosif, Ana Maria Cristina Țâncu, Andreea Cristiana Didilescu, Marina Imre, Silviu Mirel Pițuru, Ecaterina Ionescu and Viorel Jinga for publication in the IJERPH journal. We sincerely appreciate the time and effort that you dedicated to provide feedback on our manuscript and are grateful for the insightful observations and valuable improvements to our paper. Thus, we have incorporated all the suggestions which were recommended by you. Those revisions to the manuscript were marked up using the “Track Changes” Function. Please see below the point to point responses to your recommendations.

Abstract:

  1. The abstract is quite lengthy. Please summarize it with pertinent information on the psychological impact felt by academics during the COVID-19 pandemic, and the impact felt in terms of dental education in the same pandemic context, rather than listing out the entire results

R: Thank you for the suggestion, we reduced the length of the abstract, focusing mainly on the self-perceived psychological impact by academics, and the impact felt in terms of dental education.

  1. However, to the best of our knowledge, no scientific study has focused strictly on the vision of the academic body in the field of dentistry ….” This phrase may not be appropriate as currently there is much evidence, in fact several review articles have been published on this matter.

R: We removed this phrase completely, as you can see in the shortened form of the abstract.

Introduction:

  1. Nonetheless, to the best of our knowledge, there is no study described in the literature that strictly investigated the perspective of the dentistry academic body on the way the abrupt transition from physical to online…” Similarly, please revise this sentence. Perhaps, such a study may be considered novel in a particular country, such as Romania, but it is not considered novel throughout the globe as similar studies have been conducted in several countries.

R: Thank you for your suggestion, we modified accordingly and highlighted the particularity of this study for Romania.

Methodology:

  1. 93 university teachers (n=93) complied with our request, forming the present study…” Can omit the (n=93) since the authors have listed the number of teachers who participated in the study.

R: We omitted this parenthesis, as you suggested.

  1. Please summarize “2.3. Survey Instrument”.

R: We summarized the information regarding the answer direction of the scale in a single sentence and eliminated the series of questions in this part of the manuscript, considering the fact that they are also found in the Supplementary File.

  1. Did the authors perform any reliability test on the questionnaire items such as Cronbach alpha?

R:  Yes, a reliability analysis was made on the questionnaire items, for the two categories of analysed items: the psychological impact (questions 7 to 10) and the educational impact (questions 11 to 17) of the COVID-19 pandemic. A higher value for the score of the item would represent a higher impact perceived by the respondents for that item. Therefore, a recoding was necessary to be made for questions 11, 15, 16 and 17 (so that a higher agreement towards that item signified a higher educational impact). The analysis is represented in the following table:

Characteristic

Item

Cronbach’s alpha if Item is Deleted

Model

Psychological impact

Q7

0.637

0.686

Q8

0.629

Q9

0.562

Q10

0.648

Educational impact

Q11

0.746

0.767

Q12

0.749

Q13

0.723

Q14

0.740

Q15

0.743

Q16

0.737

Q17

0.726

According to the presented table the items which analysed the psychological impact had just a close to 0.7 internal consistency (α = 0.686), because of the low number of items (n=4) and because most of the consistency was attributed to the question 9 – sleep impact (α = 0.562 if Q9 is deleted). The items which analysed the educational impact had an good internal consistency (α =.767), and the model’s consistency was not strongly influenced by the deletion of one of the items (signifying a homogenous consistency of the items in analysing the perceived characteristic – the educational impact). We added now the Cronbach-alpha values for each section in the revised form of our manuscript.  

Results:

  1. I would highly suggest rewriting the results. It is advisable to categorize the results in a manner that suit the questionnaire design. The authors designed the questionnaire into three sections. Hence, present the results accordingly. For instance, the authors classify section 2 as psychological impact.

R: Totally agree, we restructured the results section according to the sections of the questionnaire, thank you.

  1. How are the overall results of the psychological impact? Is it the majority of the teachers agreed that their psychological impacts are significant due to the pandemic? Besides, the authors divided the psychological impact into emotionally affected, stress impact, sleep quality and anxiety. Which one has the most significant impact?

R:  Overall, the psychological impact imposed by the pandemic among dental teachers was rather low, although almost a third of the respondents declared themselves emotionally affected during the pandemic, most rejected the impact of personal stress on academic performance, denied the decrease in sleep quality or the increase in the feeling of anxiety. This finding, as well as the one from the educational section, was reformulated more clearly in the abstract and highlighted more in the results and discussion part of the manuscript.

  1. The readers can understand based on the tables and thus, the authors should summarize the results instead of repeating what is written in the tables.

R: We eliminated all the redundant information that was found next to the tables and summarized the results, thank you for your suggestion.

  1. There are lots of tables which some of them can be merged.

R: Indeed, we combined tables 4-7 into a single table.

Discussion:

  1. Within the university medical education in all countries around the world it has generated …family life, as well as on their professional prospects, which have been severely affected by the prolonged lockdown.” Please at least put a reference to these statements.

R: Thank you very much for reporting this, we reworded the entire paragraph inserting the appropriate references

  1. Some of the discussion points are not relevant to the findings. The discussion should be confine to what happened to the findings? What factors could lead to the present findings? Are the findings in accordance or contradicts previously similar studies? What recommendations the authors would like to propose?

R: Thank you for pointing this out. Thus, we have completely organized the discussion part, according to the sequence of the questionnaire and the significant findings.

Thank you very much for your kind and insightful assistance!

Reviewer 2 Report

The authors take up a topic in their manuscript that has been the subject of many studies, so in order to emphasize the importance of the topic taken, I recommend that the manuscript title be modified in terms of originality. I also recommend that the authors refine this element in abstract. It should be emphasized primarily in order to encourage the reader to read the work.

The abstract needs some refinement. The authors do not specify the period in which the study was conducted. In the abstract, the goal of the work is not clear. I recommend to improved. The conclusions given in the abstract are nothing innovative, so I recommend substantively emphasizing why the manuscript is important and what it brings to science. 

Introduction. In line 119 authors write that: "A small number of studies have focused on the perceptions of university teachers on" I recommend that you refer to the literature that the authors mention in order to support the statement. I recommend also that the authors justify the importance of the topic in the introduction.

Materials and Methods. In line 136 authors write that: "...degree of impact perceived by them..." this sentence is not clear. 

In line 137 in recommend to put exactly date of realizing the survey.

In line 139 authors write that: "Faculty of Dentistry (n═320; females═207, males═113)" so I suggest that in line 142: "93 university teachers (n=93) " also add information  about females and males number.

2.2. Data collection and ethical considerations

The authors write in line 145, "...three members of the Faculty of Dentistry were distributed...", while in point 2. 1. Study design and sampling procedures in line 134 authors mention 2 universities: the Faculty of Dentistry of the "Carol Davila" University of Medicine and Pharmacy in Bucharest, Romania. Its chaotic. I recommend to organize this informations. 

The strong point of the work is an interesting topic and methodology. The weak point, however, is the lack point literature review. I recommend that the authors create a Literature review section and insert it before the Materials and Methods section.

The weakness of the manuscript is the discussion too. First of all, this point is written very chaotically. I have the impression that the authors wrote this point without any plan. When reading the discussion, the authors do not refer to literature, moreover, many sentences are obvious and do not add anything to the manuscript. I recommend removing such sentences in lines 411-420:

"The concerns of dental academics about the major vulnerabilities experienced by dental students have been equally about  psychological and educational vulnerabilities, with almost every state in the world today  (to the best of our knowledge), having submitted relevant scientific data on the effects of this calamity on the students' physical and mental health, on their social, financial and family life, as well as on their professional prospects, which have been severely affected by the prolonged lockdown. The most popular methods used to try to quantify the severity of this impact among the younger generation in dental academia were also in line  with the switch from physical to online teaching mandated by the pandemic, namely the  application of web or internet surveys."

The authors provide in lines 420-427 the research results and percentages, which I propose to present graphically. The sentence is written so intricately that it is not clear what these percentages mean. In line 426 authors write: "Although we would have expected that this issue would find..." again, it's not clear what the issue is. I believe that the conclusion point should be completely redone. The authors refer to results, which means that part of the text should be moved to the results section. In the discussion, reference should be made to the research and literature on the perception of University Teachers towards the COVID-19, taking into account the challenges currently faced by medical universities. Conclusion is a very soft point. This section needs elaboration.  The conclusion point needs to be refined, especially in relation to the purpose of the work, the research carried out, but also the literature.

The authors in the work should also demonstrate extensive knowledge of current literature, which I lacked after reading the work. 

Author Response

Dear Reviewer,

Thank you for giving us the great support to submit a revised form of the manuscript firstly entitled PERCEPTIONS AND EXPECTATIONS OF UNIVERSITY TEACHERS TOWARDS THE COVID-19 PANDEMIC IMPACT ON DENTAL EDUCATION authored by Laura Iosif, Ana Maria Cristina Țâncu, Andreea Cristiana Didilescu, Marina Imre, Silviu Mirel Pițuru, Ecaterina Ionescu and Viorel Jinga for publication in the International Journal of Environmental Research and Public Health.  According to your well emphasized comments and suggestions, we tried to do our very best to remove all the ambiguities which might have occurred. Those revisions to the manuscript were marked up using the “Track Changes” Function. Please see below the point to point responses to your recommendations.

  1. The authors take up a topic in their manuscript that has been the subject of many studies, so in order to emphasize the importance of the topic taken, I recommend that the manuscript title be modified in terms of originality.

R: We have modified the title so that the originality and importance of the topic addressed are emphasized, thank you very much for your pertinent observation.

  1. I also recommend that the authors refine this element in abstract. It should be emphasized primarily in order to encourage the reader to read the work.

R: Thank you for your kind recommendation, we emphasized this element in the current, shortened version of the abstract.

  1. The abstract needs some refinement. The authors do not specify the period in which the study was conducted. In the abstract, the goal of the work is not clear. I recommend to improved. The conclusions given in the abstract are nothing innovative, so I recommend substantively emphasizing why the manuscript is important and what it brings to science. 

R: According to your suggestions we introduced the period of the study in the abstract and highlighted more the objectives, importance and conclusions of our study, thank you very much.

  1. Introduction. In line 119 authors write that: "A small number of studies have focused on the perceptions of university teachers on" I recommend that you refer to the literature that the authors mention in order to support the statement. I recommend also that the authors justify the importance of the topic in the introduction.

R: We redid the entire paragraph in accordance with your suggestion, with reference to several scientific  studies.

  1. Materials and Methods. In line 136 authors write that: "...degree of impact perceived by them..." this sentence is not clear. 

R: Indeed, we have modified the wording for more clarity, thank you for your observation.

  1. In line 137 in recommend to put exactly date of realizing the survey.

R: The date of application of the questionnaire can be found a little below in the paragraph, there are two mentions in this regard, thank you.

  1. In line 139 authors write that: "Faculty of Dentistry (n═320; females═207, males═113)" so I suggest that in line 142: "93 university teachers (n=93) " also add information about females and males number.

R: We added the information about the number of women and men participating, as you suggested.

8- 2.2. Data collection and ethical considerations

The authors write in line 145, "...three members of the Faculty of Dentistry were distributed...", while in point 2. 1. Study design and sampling procedures in line 134 authors mention 2 universities: the Faculty of Dentistry of the "Carol Davila" University of Medicine and Pharmacy in Bucharest, Romania. Its chaotic. I recommend to organize this informations. 

R: It is about the same faculty, that of Dentistry, which is part of the University of Medicine and Pharmacy in Bucharest, along with the Faculty of General Medicine, the Faculty of Pharmacy and respectively the Faculty of Midwifery and Nursing. We have properly organized the information so that there are no more confusions, thank you for your kind observation.

  1. The strong point of the work is an interesting topic and methodology. The weak point, however, is the lack point literature review. I recommend that the authors create a Literature review section and insert it before the Materials and Methods section.

R: True, that's why in the introduction we included a series of specialized references, regarding studies that evaluated the impact felt in general by the university environment, then in the medical and dental respectively.

  1. The weakness of the manuscript is the discussion too. First of all, this point is written very chaotically. I have the impression that the authors wrote this point without any plan.

R: Thank you for pointing this out. Thus, we have completely organized the discussion part, according to the sequence of the questions of the survey and the significant results.

  1. When reading the discussion, the authors do not refer to literature, moreover, many sentences are obvious and do not add anything to the manuscript. I recommend removing such sentences in lines 411-420:

"The concerns of dental academics about the major vulnerabilities experienced by dental students have been equally about  psychological and educational vulnerabilities, with almost every state in the world today  (to the best of our knowledge), having submitted relevant scientific data on the effects of this calamity on the students' physical and mental health, on their social, financial and family life, as well as on their professional prospects, which have been severely affected by the prolonged lockdown. The most popular methods used to try to quantify the severity of this impact among the younger generation in dental academia were also in line with the switch from physical to online teaching mandated by the pandemic, namely the  application of web or internet surveys."

R: Thank you for your observation, thus we reformulated and shortened the entire paragraph accordingly, adding all the cited bibliographic resources.

  1. The authors provide in lines 420-427 the research results and percentages, which I propose to present graphically. The sentence is written so intricately that it is not clear what these percentages mean. 

R: The paragraph refers to online questionnaires applied to dental students worldwide, not talking about the results of the questionnaire applied to the academic staff in our study, as follows:

The response rate among dental students to these questionnaires was generally variable, reaching for example a high of 80% for the University of Giessen, Germany [6], 90.72% for the only dental school in Malta [7], 72% for the University of Jordan [8], but also low, as reported for example by the University of Washington, USA, of 35.5% [9], or moderate, such as that of the Faculty of Dental Medicine in Vienna, Austria, of 47% [10], and by our faculty in Bucharest, Romania with a percentage of 48.56% [11].

If you still consider it appropriate to create a graph in this situation, please let us know.

  1. In line 426 authors write: "Although we would have expected that this issue would find..." again, it's not clear what the issue is.

R: We removed the term issue and replaced it as follows, for better clarity of the text:

Although we would have expected that the problem of dental education in the era of COVID-19 would find a wider response among university teachers ….

  1. I believe that the conclusion point should be completely redone. The authors refer to results, which means that part of the text should be moved to the results section. In the discussion, reference should be made to the research and literature on the perception of University Teachers towards the COVID-19, taking into account the challenges currently faced by medical universities. Conclusion is a very soft point. This section needs elaboration. The conclusion point needs to be refined, especially in relation to the purpose of the work, the research carried out, but also the literature.

R: Indeed, we reformulated the conclusions at your suggestion and highlighted the consequences of these results for dentistry education in Romania and what solutions and optimization perspectives for this important segment of medical education await for the future.

  1. The authors in the work should also demonstrate extensive knowledge of current literature, which I lacked after reading the work. 

R: Your observation helped us a lot to improve the discussion part as well as the introduction with much more data and comparative references from the current literature, thank you. 

Thank you very much for your kind assistance!

Round 2

Reviewer 1 Report

Thank you for revising the manuscript. It looks far better than the first version. 

However, before it can be deemed suitable to be accepted, please remove parts of the discussion. A research study's discussion should be concise and informative. The readers are not expecting to read a 'thesis' version of manuscript. 

For example, is the reason of respondent rate very significant? I do agree that discussing the respondent rate is acceptable but it should not be the whole paragraph about respondent rate. The readers want to know 'why' and 'how' regarding the main outcomes. Same goes to gender. Those are important but are not the main objective of your study.

Besides, some citations are not necessary. For instance:

"Our 423 findings are in agreement with Martin F. et al. [53] and Cassachia M. et al. [54], who rec- 424 orded the same challenges in terms of technology and digital learning on the other side of 425 the barricade, namely among female students [55, 56]." You can remove 55 and 56 since you have quoted 53 & 54. 

*Revise the discussion only*

Author Response

Dear Reviewer,

Thank you for giving us the support to resubmit a revised form of the manuscript firstly entitled PERCEPTIONS AND EXPECTATIONS OF ACADEMIC STAFF IN BUCHAREST TOWARDS THE COVID-19 PANDEMIC IMPACT ON DENTAL EDUCATION authored by Laura Iosif, Ana Maria Cristina Țâncu, Andreea Cristiana Didilescu, Marina Imre, Silviu Mirel Pițuru, Ecaterina Ionescu and Viorel Jinga for publication in the International Journal of Environmental Research and Public Health.  Please see below the point to point responses to your recommendations.

Comment 1: Please remove parts of the discussion. A research study's discussion should be concise and informative. The readers are not expecting to read a 'thesis' version of manuscript.

For example, is the reason of respondent rate very significant? I do agree that discussing the respondent rate is acceptable but it should not be the whole paragraph about respondent rate. The readers want to know 'why' and 'how' regarding the main outcomes. Same goes to gender. Those are important but are not the main objective of your study.

Response: Thank you very much for your kind recommendation. Although we agree with the extended space that we dedicated to the response rate of the respondents, we would still like to keep all the explanations that we consider plausible on this topic, both the reports of previous studies and everything that concerns the internal situation in Bucharest, at the time of application of the questionnaire. In our opinion, this is an important feature that our study presents to the readers of this manuscript.

Totally agree with the paragraph on gender, we removed lines 353-355.

The discussion part of the manuscript is considerably shortened in the present version compared to the first submitted version, strictly containing comments related to the significant results obtained.

Comment 2: Besides, some citations are not necessary. For instance:

"Our 423 findings are in agreement with Martin F. et al. [53] and Cassachia M. et al. [54], who rec- 424 orded the same challenges in terms of technology and digital learning on the other side of 425 the barricade, namely among female students [55, 56]." You can remove 55 and 56 since you have quoted 53 & 54.

Response: Thank you for your observation, indeed, we removed the redundant references 55 and 56.

Thank you very much for your kind assistance!

Reviewer 2 Report

Thank you very much for including all the recommendations in the manuscript. Thank you for your effort. The manuscript looks very good, is readable, and is attractive to the reader. Congratulations to the authors.

Author Response

We thank you very much for your comments and kind assistance!